# Zooplankton impact on lipid biomarkers in water column vs. surface sediments of the stratified Eastern Gotland Basin (Central Baltic Sea)

**Anna K. Wittenborn**[1¤]*, **Oliver Schmale**[2], **Volker Thiel**[1]

**1** Geoscience Center, Georg-August University of Göttingen, Göttingen, Germany, **2** Marine Chemistry, Leibniz Institute for Baltic Sea Research Warnemünde, Rostock, Germany

¤ Current address: Marine Geology, Leibniz Institute for Baltic Sea Research Warnemünde, Rostock, Germany

\* anna.wittenborn@io-warnemuende.de

**Data Availability Statement:** All relevant data are within the manuscript and its Supporting Information files.

## Abstract

Sediments from stratified marine environments often show an enhanced preservation of organic matter (OM) which is attributed to the limitation, or absence, of oxygen in the bottom waters and surface sediments. Yet there is still a limited knowledge about the changes that the associated biomarker signals undergo in the different parts of a stratified environment, and as to which extent the situation in the productive upper parts of the water column is eventually reflected in the sedimentary record. To better understand these processes we studied particulate matter samples from the stratified, partly anoxic Eastern Gotland Basin (EGB, Central Baltic Sea) during a strong cyanobacterial bloom in August 2016. Endmember samples representing the main biomass producers within the phytoplankton (cyanobacteria) and mesozooplankton (copepods) were obtained from different levels of the water column. Major extractable lipids (fatty acids, n-alcohols, sterols, and selected hydrocarbons) were analysed from the same materials and compared to samples cored from the underlying surface sediments (0–12 cm). Given the annually recurring phenomenon of cyanobacterial blooms we anticipated to find a considerable lipid footprint of the major primary producers in the sedimentary record of the EGB. Unexpectedly, however, lipids in the surface sediments largely derived from the storage lipids (mainly wax esters) of the copepod *Pseudocalanus* spp. which thrived in deeper, more saline and oxygen-depleted waters. Carbon number and unsaturation patterns suggest that the component n-alcohols of these wax esters are transformed into the corresponding n-fatty acids prior to further degradation in the sediment. In the EGB deposits, most of the plankton-derived lipids studied appear to be degraded on a time scale of decades. In terms of relative abundances, long-chain n-alkyl lipids and $C_{29}$ sterols from terrestrial plant sources instead become predominant in the deeper sediment layers. Likewise, higher stanol/sterol ratios of $C_{27}$-sterols vs. $C_{29}$-sterols indicate a more intense biodegradation of planktonic OM as compared to terrestrial OM. Our observations indicate that primary produced particulate OM is heavily modified by mesozooplankton grazing. This overprint adds on the influence of heterotrophic microorganisms and, in the

**Funding:** The work was funded by the grants of the German Research Foundation (SCHM 2530/5-1) awarded to OS and grant TH713/12-1 awarded to VT. The publication of this article was funded by the Open Access Fund of the Leibniz Association. The funders had no role in study design, data collection and analysis, decision to publish, or preparation of the manuscript.

**Competing interests:** The authors have declared that no competing interests exist.

sediment, preferential preservation of terrestrial biomarkers. Taken together, these factors result in a major decoupling of the biomarker signals between the productive upper mixed layer and the oxygen-depleted bottom waters and sediments of the EGB.

## 1. Introduction

Stratified aquatic environments showing oxygen depletion in the water column promote the input of organic carbon into the geosphere, due to a restricted heterotrophic consumption of primary produced organic matter (OM). Consequently, stratification is commonly regarded as an important prerequisite for the deposition of organic-rich sediments, including black shales and hydrocarbon source rocks in the geological past [e.g. 1, 2]. However, the controls on OM deposition and enrichment in stratified settings are still under research and it is not fully established as to which extent the biogeochemistry and ecological situation in the overlying stratified water column is actually reflected in the sedimentary record [e.g. 2, 3–5].

Available data suggest that the boundary between well mixed oxic surface waters and more reducing bottom waters ('chemocline') represents a potential hotspot of biological activity hosting very active prokaryotic and eukaryotic communities. These biota directly or indirectly utilize the steep biogeochemical gradients that typically occur in such settings (e.g., methanotrophs, sulfide oxidizers, grazing protozoans) [6]. Consequently, chemoclines may have a considerable impact on the OM deposition in that they represent (i) a site of intense degradation of surface-derived OM, and (ii) a 'factory' of additional microbial OM that would eventually add to the sedimentary OM pool. In fact, chemocline-derived organic compounds have been clearly identified in stratified water columns [7, 8] and their underlying sediments [7, 9]. However, their overall contribution to the sedimentary OM pool is as yet unknown.

Another major factor influencing the sinking primary produced OM in most aquatic settings is heterotrophic grazing, particularly by copepods (which represent > 50% of the mesozooplankton biomass in most areas of the Baltic Sea, including the Eastern Gotland Basin (EGB); [10], their Fig 3). Zooplankton behaviour (e.g. selective and sloppy feeding, vertical migration) and fecal pellet production affect the fate of primary produced OM throughout the water column [11, 12]. Overall, these factors clearly have the potential to significantly alter, or even eliminate the primary OM signal before it is ultimately deposited. Therefore, information on how these processes control the inventory of sedimentary organic compounds is crucial for a sound interpretation of the biomarker record from modern and ancient stratified ecosystems.

The Baltic Sea is an ideal site for studies on the OM dynamics associated with stratification. Linked to the North Sea only via the narrow Skagerrak/Kattegat straits, it is a more or less landlocked brackish basin comprised of several sub-basins, with the Eastern Gotland Basin (EGB) being the largest (Fig 1). The EGB has a maximum water depth of 248 m. A seasonal thermocline is regularly established at 10–30 m during summer, whereas a permanent chemocline exists at 80–130m [13]. Below the chemocline, waters are persistently oxygen-depleted and may even become periodically anoxic, unless episodic inflows of more saline waters from the North Sea lead to a temporary replenishment of oxic conditions [14].

The current environmental situation of the Baltic Sea may provide useful clues for the reconstructions of ancient settings. Some facets, e.g. its deeply structured hydrographic configuration, terrestrial influence (sediments, nutrients, freshwater inflow), and stratification with intermittent anoxicity of deeper water layers, as seen in the EGB, [16] are comparable to

similar marginal- or restricted marine settings in the geological past. A suitable example may be the epicontinental Sea that gave rise to the deposition of the organic-rich Posidonia Shale in mid-Europe during the Early Jurassic [1]. Further, the question as to which extent cyanobacteria comprised an important factor in primary production is a long-standing problem in the reconstruction of ancient ecosystems [4, 17].

Here we report on organic compounds in particulate matter samples collected from the EGB in August 2016 during a cyanobacterial bloom dominated by the filamentous, nitrogen-fixing specie *Nodularia spumigena* [18]. Particulate OM, including end member samples of the relevant phyto- and mesozooplankton, was retrieved from different layers of the stratified water column. The samples were studied by microscopy for the composition of key species, and analysed for major lipid biomarkers (fatty acids, n-alcohols, sterols, and selected hydrocarbons). Comparison with the immediately underlying sediments provides an insight into the consequences of zooplankton grazing and other heterotrophic processes on the sedimentary biomarker signal left by major primary producers (here: cyanobacteria) in the stratified waters of the Central Baltic Sea.

## 2. Materials and methods

### 2.1. Sampling

Field work was conducted during cruise AL483 of R/V *Alkor* in August 2016. Samples were retrieved at station TF 0271 in the central EGB (Fig 1; 57 19.2209' N/ 20˚03.0153'E; water depth 248 m). Field site access was granted by Latvian Ministry of Foreign Affairs. Physico-chemical parameters of the water column including temperature, oxygen, salinity and turbidity were examined using a SBE 911 CTD system (Seabird Electronics, USA). In summer 2016, the EGB exhibited a stable stratified water column with a pronounced thermocline located at 20-25m and a chemocline separating the oxic surface waters from oxygen-depleted bottom waters at 60-75m (for details, see 3.1 and Fig 2). Phytoplankton and mesozooplankton distributions were observed by light microscopy.

(i) Bulk phytoplankton (0–25 m), largely comprised of cyanobacteria, was collected with an Apstein-net (mesh size 50 μm). Separation from co-sampled mesozooplankton was achieved using a simple self-built trap consisting of a 1.5 L transparent tapped plastic bottle that was filled with the sample water and partly wrapped into a black non-transparent bin bag. Meso-zooplankton was attracted by a light source at the uncovered bottom of the bottle, and drained off after a few hours. An aliquot of the remaining phytoplankton was sieved and fixed for light microscopy with lugol [19]. This sample will hereafter be referred to as 'phytoplankton (0–25 m)'.

(ii) For a homogenous sampling of mesozooplankton advantage could be taken from the diurnal migration behaviour of copepods. These animals leave the light-penetrated upper mixed layer during the daytime and migrate towards greater depths to avoid grazing, which was monitored by echo sounding [20]. Therefore sampling was performed during the daytime in the 25–60 m depth interval (cold winter layer) using a WP-2 net (mesh size 100 μm, opening 57 cm, see also [19, 20]. This sample will hereafter be referred to as 'zooplankton (25–60 m)'.

(iii) Furthermore, a haul was performed with the setup described in (ii) in the upper oxygen-depleted zone (60–90 m). The catches were concentrated with a hand sieve (20 μm). Aliquots for microscopy were fixed with borax-buffered formaldehyde [19]. This sample will hereafter be referred to as 'zooplankton (60–90 m)'.

(iv) Small organisms thriving at the chemocline (largely ciliates and chemoautotrophs) were collected using eight 10 l Niskin bottles that were closed in 5 m intervals in the upper oxygen-depleted zone (60–95 m). The water was pooled, coarsely pre-filtered using a sieve

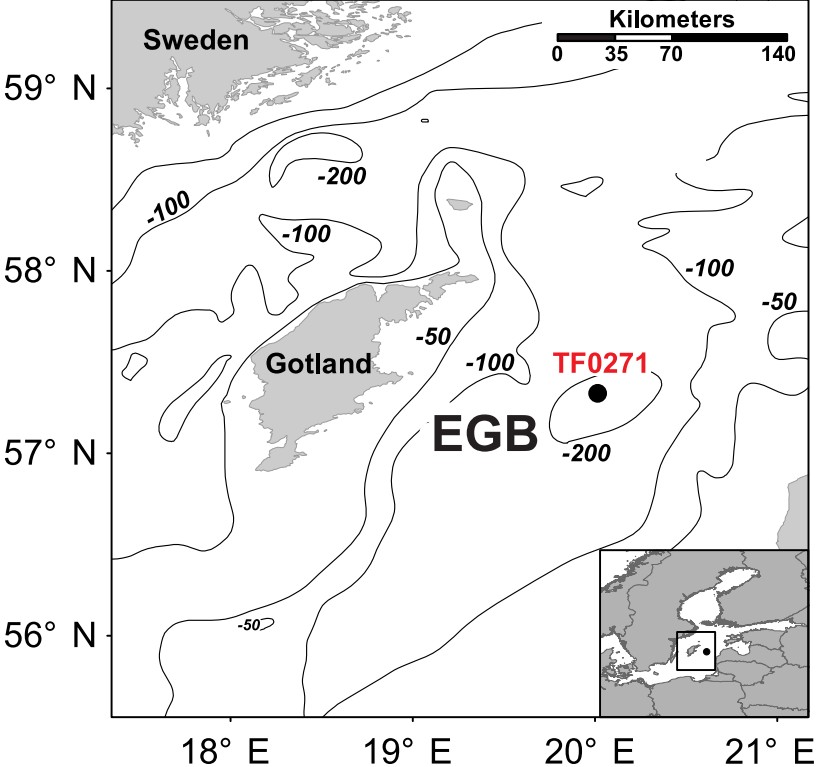

**Fig 1. Map of the study area.** The black dot marks the location of sampling station TF0271 in the central part of the Eastern Gotland Basin (Map data based on [15]).

(100 μm) and passed through pre-combusted GFF-filters (diameter 47 mm, mesh size 0.7 μm). This sample will hereafter be referred to as 'filter (60–95 m)'.

(v) For sediment sampling, a core was retrieved at the same station using a Frahmlot, and nine sediment samples were obtained between 0–12 cm at a depth of ~270m. These samples will hereafter be referred to as 'sediment (x-y cm)'. Published age data for similarly laminated sediment cores retrieved near our sampling location in the east EGB [21] indicate an age of approximately 30 years for the deepest sediment layer studied (10–12 cm).

All samples for lipid biomarker analysis were frozen at -20°C until further analyses in the home laboratory.

## 2.2. Organic carbon analysis

Total organic carbon ($C_{org}$) was analyzed using a LECO RC-612 multiphase carbon analyzer. $C_{org}$ values were calculated by comparison of the sample peak areas with those of certified standards (LECO).

## 2.3. Biomarker analysis

**2.3.1. Extraction of biomarkers.** About 25 mg each of the lyophilized samples were extracted 3 x with each 10 ml portions (filter sample: 50 mL) of dichloromethane (DCM)/ methanol (MeOH) (2:1, v:v), DCM/MeOH (3:1, v:v) and DCM/n-hexane (2:1, v:v) using ultra-sonication (15 min, 20°C). After each extraction step, the samples were centrifuged, and the extracts combined into a total organic extract (TOE). For fatty acid (FA) analysis, an aliquot of the TOE was separated into neutral lipids (NL, largely containing storage lipids and sterols)

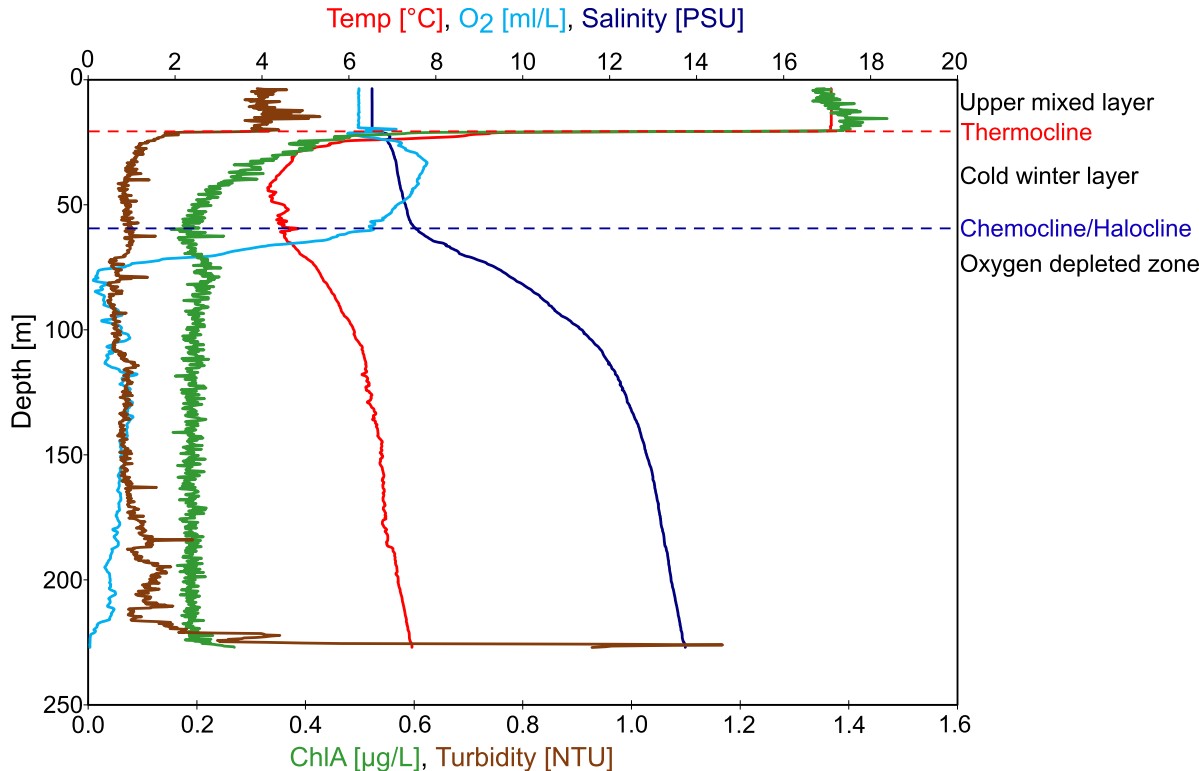

**Fig 2. CTD-profile showing physico-chemical parameters of the water column (EGB, station TF0271, 14 August 2016, cruise AL483).**

and polar lipids (containing polar cell membrane lipids including phospholipid fatty acids, PLFA) using column chromatography. A glass column (internal diameter 13 mm), containing a frit and 0.7 g silica gel (60–230 μm, thickness of the silica bed 10 mm) was conditioned with n-hexane. The TOE was vapor-deposited onto a small amount of silica gel and applied to the column. The NL fraction was eluted with 7 ml DCM/acetone (9:1, v:v) and the PLFA fraction with 7 ml DCM/MeOH (3:1, v:v).

**2.3.2. Derivatization of fatty acids.** To enable gas chromatographic analyses, FA were cleaved from the parent molecules (triglycerides, wax esters, phospholipids) and converted to their methyl ester derivatives (FAME) by addition of trimethylchlorosilane (TMCS) in MeOH (1:9, v:v; 90 min, 80˚C). The resulting FAME were recovered from the reaction mixture by liquid-liquid-extraction with 1 ml portions of n-hexane (3x). The combined extracts were carefully evaporated with nitrogen to near-dryness, and further derivatized using a mixture of 100 μl n-hexane and 100 μl BSTFA/pyridine (3:2, v:v) to convert alcohols to their trimethylsilyl (TMS-) derivatives (40˚C for 60 min). The resulting fractions were then subjected to gas chromatography–mass spectrometry (GC-MS) analysis.

**2.3.3. Gas chromatography–mass spectrometry (GC-MS).** Compounds were analysed with a Thermo Fisher Trace 1310 GC coupled to a Thermo Fisher Quantum XLS Ultra MS. Fractions were injected with an autosampler (Thermo TriPlus RSH) onto a fused-silica capillary column (Phenomenex Zebron ZB-5MS, 30 m, 0.1 μm film thickness, inner diameter 0,25 mm). The carrier gas was He at a flow rate of 1.5 mL/min. The GC oven temperature was ramped from 80˚C (1 min) to 310˚C at 5˚C/min and held for 20 min. Electron ionization mass spectra were recorded in full scan mode at an electron energy of 70 eV within a mass range of m/z 50 to 600.

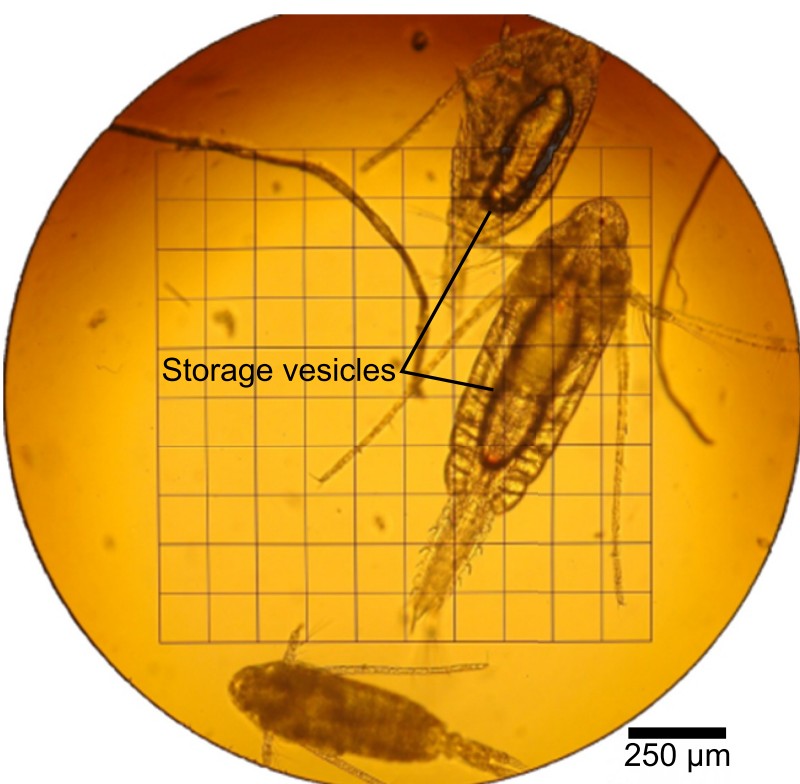

**Fig 3. Microscopic image of the deep water copepod Pseudocalanus spp.** *Pseudocalanus* spp. dominates the zooplankton in the upper oxygen-depleted zone (60–90 m). Note the large lipid-filled storage vesicles in the interior of the animals.

Compounds were identified by their retention times and comparison with published mass spectra. FAME were characterized by comparison of mass spectra and retention times with an external standard mixture (Supelco 37 FAME standard). Compounds were quantified by comparison with internal standards of known concentration.

**2.3.4. Miscellaneous.** All glassware (pipettes, vials, beakers) was heated to 400˚C for 2 h. Solvent rinsing (deionized water and acetone) was carried out on steelware (spatula, tweezers, saw blade) prior to use.

## 3. Results

### 3.1. Physicochemical parameters of the water column

During the sampling time in august 2016 the EGB showed a pronounced vertical stratification (Fig 2) that roughly divided the water column into three layers, as follows.

(i) The **upper mixed layer** (0–25 m; syn. 'warm surface layer', [8] was warm (T = 17˚C), fully oxygenated ($O_2$ = ~6 mL/L), low in salinity (6 PSU), and was the main site of primary production (Chl a = ~1.4 μg/L).

(ii) The **cold winter layer** (25–60 m) (T = 5˚C) was separated from the upper mixed layer by a pronounced thermocline. It was influenced by winter convection and was thus slightly higher in oxygen ($O_2$ = ~7 mL/L), at similarly low salinity (~ 7 PSU). Primary production dropped sharply in this layer due to low light intensities (Chl a = ~0.3 μg/L).

Together, (i) and (ii) represent the **'oxic zone'** of the water column (syn. 'above chemocline').

(iii) The **oxygen-depleted zone** (60–248 m; syn. 'below chemocline', bottom layer', 'suboxic zone') is separated from the cold winter layer by a **chemocline** (60 – 75m, syn. redoxcline, halocline) and is characterized by relatively stable temperatures (~5˚C, slightly increasing with depth), and a major drop in oxygen concentrations ($O_2$ = <1 mL/L). Salinity is increasing towards the bottom (~15 PSU) whereas markers for primary production stay at low concentration (Chl a = ~0.3 μg/L).

It should be noted that, in August 2016, the oxygen content in the oxygen-depleted zone showed considerable variation. First, a drop from high values of the cold winter layer to near-zero values was observed in the region of the chemocline (60–75 m). Below that depth, $O_2$ increased again to around 1 mL/L before it dropped to zero in the near-bottom layer at about 225 m (Fig 2). This variation can be explained by inflows of different oxygen-containing water masses [22].

## 3.2. Microscopy

In the phytoplankton (0–25 m) sample, cyanobacteria contributed most (> 95%) of the biomass observed. *Nodularia spumigena* (Nostocophyceae) accounted for approximately 65% and *Aphanizomenon* sp. for *35%* of the total cyanobacteria. In august 2016, the mean biomass of *Nodularia spumigena* recorded in the EGB was 122 μg/L [23].

The most abundant specie in the zooplankton (25–60 m) sample from the cold winter layer was the copepod *Temora longicornis* (52% of the mesozooplankton). In addition, *Acartia* spp. (20%) and *Centropages* sp. (20%) were moderately abundant. Low counts were furthermore observed for *Pseudocalanus* spp. (4%) and *Eurytemora* sp. (3%).

In contrast, the zooplankton (60–90 m) sample from the upper oxygen-depleted zone showed *Pseudocalanus* spp. as the main specie by far (58%). *Temora longicornis* accounted for 19% whereas *Centropages* sp. and *Acartia* spp. each constituted around 10% of the mesozooplankton.

It should be noted that the individual numbers and biomass of copepods in surface waters are subject to major diurnal variation due to the migration behaviour of these animals. The total abundance of copepods (integrated between surface and halocline) during the sampling time at station TF 0271 was 19–25 ind./L [19]. For comparison, mean abundances of copepods for the Mecklenburg Bay and the Arkona Basin in summer 2016 were in the range of 12–15 and 7–15 ind./L, respectively [23]. Similar values were also reported for the adjacent Gdansk Basin in summers 2010 and 2011 (4–18 ind./L; according to a biomass of 92 and 676μg/L, respectively; [24]).

## 3.3. Fatty acids (FA)

**3.3.1. FA in the water column.** The phytoplankton (0-25m) contained 41 mg lipids $g^{-1}$ $C_{org}$ whereas the zooplankton (25-60m) contained 83 mg lipids $g^{-1}$ $C_{org}$. The zooplankton (60–90 m) from the upper oxygen-depleted zone yielded an extremely high amount of lipids (408 mg $g^{-1}$ $C_{org}$), obviously due to the presence of large storage lipid vesicles in the dominant copepod *Pseudocalanus* spp. (Fig 3). The NL and PLFA distribution of planktonic samples is displayed in Fig 4. The distributions of main FA in the NL fractions are displayed in Fig 5. Detailed concentrations of major FA in the NL and PLFA fractions are given in S1 Table.

Samples from the oxic zone showed an overall similar FA distribution pattern. Interestingly, 18:3ω3 was abundant (≈ 10% of the total FA) in the oxic zone but was virtually absent below. 16:0 and 18:0 dominated the Filter sample from the oxygen-depleted zone (60–95 m), whereas 18:1ω9c was most prominent (≈ 50%) in the zooplankton sample (60–90 m) dominated by *Pseudocalanus* spp.

**3.3.2. FA in the sediments.** The sedimentary lipid concentration decreased from the topmost sediment (17 mg $g^{-1}$ $C_{org}$) towards depth (10–12 cm, 1 mg $g^{-1}$ $C_{org}$). Detailed

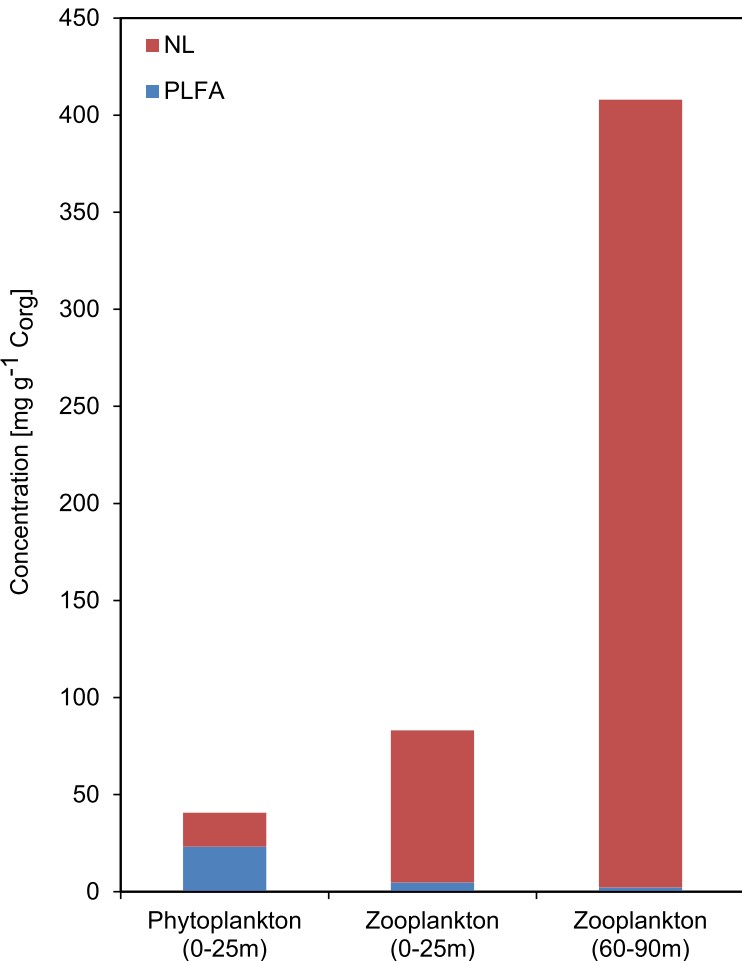

**Fig 4. Concentrations of neutral lipids (NL) and polar lipid fatty acids (PLFA) in the plankton samples.** Note the high proportion of NL in the deep water zooplankton due to abundant copepod storage lipids (compare Fig 3). NL include fatty acids and alcohols, but not sterols.

concentrations of FA ($> 1\%$) in the sediments are given in S2 Table. The main sedimentary FA ($> 5\%$) which together account for more than 80% of the total FA are shown in Fig 6. The topmost sediment (0–1 cm) showed a very high relative abundance of 16:0 and 18:1ω9c ($\approx$ 25% each). At 1–2 cm, 18:1ω9c comprised even $\approx$ 30% of the total FA, but quickly decreased to $\approx$ 10% in the 3–4 cm sample, and leveled at $\approx$ 5% below. Unexpectedly, a very-short-chain FA (14:0) made up a significant portion of the sedimentary FA ($> 10\%$), whereas its proportion was much lower ($< 5\%$) in the water column samples (see above). Moreover, the relative abundance of long-chain saturated $n$-FA increased rapidly from 9% (1–2 cm) to 27% (3–4 cm). In the deepest sample (10–12 cm) these compounds accounted for $\approx$50% of the total FA (Fig 6).

### 3.4. Sterols and tetrahymanol

**3.4.1. Sterols in the water column.** The absolute concentrations of sterols in the water column samples are given in Fig 7 and S3 Table. A notably high concentration of sterols occurred in the zooplankton sample (25-60m) from the cold winter layer ($\approx$ 15 mg g$^{-1}$ C$_{org}$). The phytoplankton sample (0-25m) also contained high amounts of sterols ($\approx$ 4 mg g$^{-1}$ C$_{org}$).

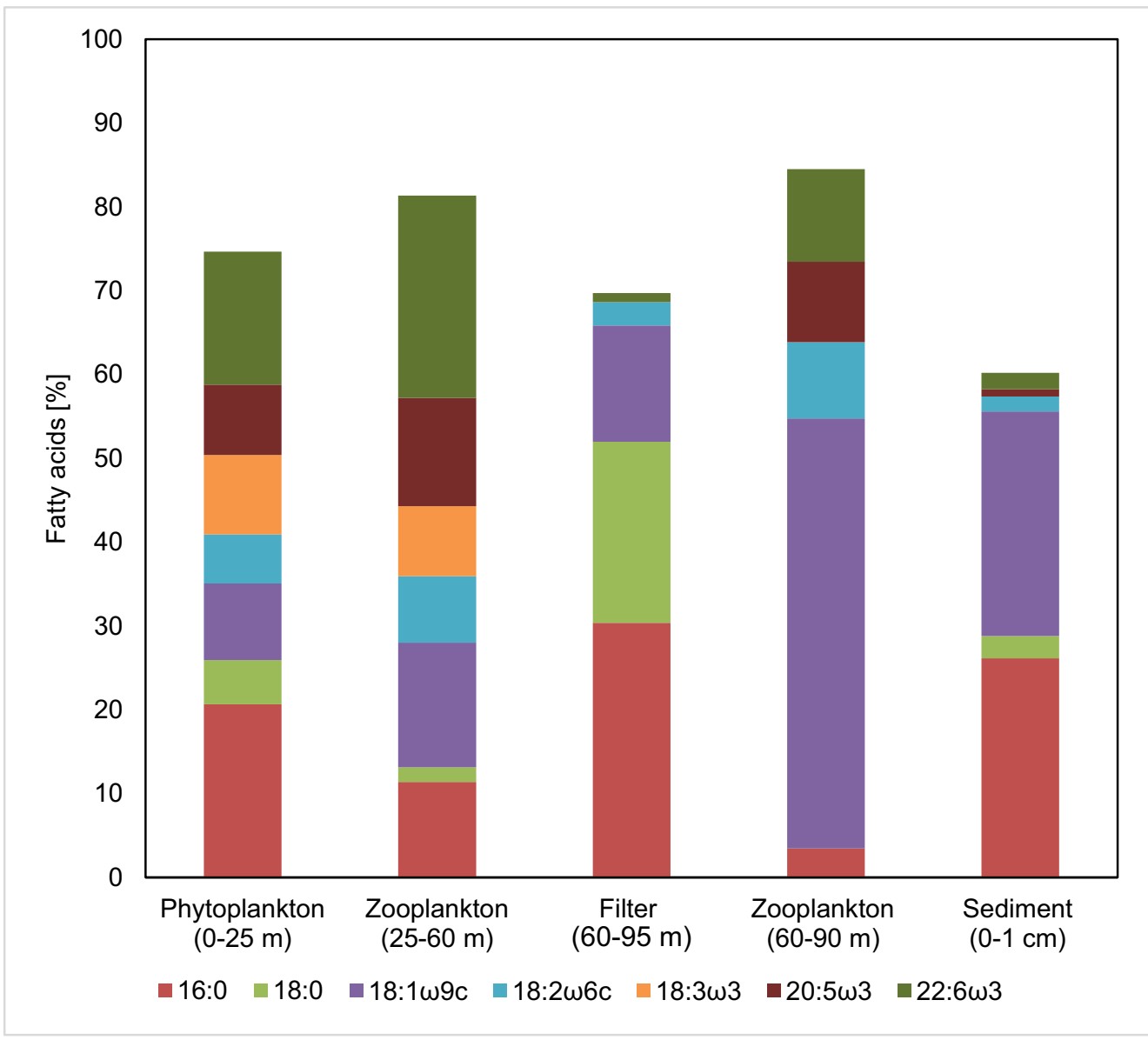

**Fig 5. Major fatty acids (FA) in the neutral lipid (NL) fractions of the water column samples.** Only compounds making up > 5% of the total in at least one sample are shown. The distribution of the total FA in the topmost sediment (0–1 cm) is shown for comparison. See S1 Table for detailed values.

Sterols in the water column consisted virtually exclusively of stenols (unsaturated sterols) whereas stanols (saturated sterols) were absent. Cholest-5-en-3β-ol (cholesterol) was the major sterol by far in all samples.

**3.4.2. Sterols and tetrahymanol in the sediments.** The absolute concentrations of sedimentary sterols are given in Fig 8 and S4 Table. Both, stenols and stanols were highest in the 0–1 cm sample (2.7 and 1.0 mg g$^{-1}$ C$_{org}$) and decreased to 335 and 155 µg g$^{-1}$ C$_{org}$, respectively, in the 10–12 cm sample. Major stenols in all sediment samples were cholesterol, 24-ethylcholest-5-en-3β-ol (sitosterol), 4α,23,24-trimethyl-5α-cholest-22E-en-3β-ol (dinosterol; Fig 8). The major stanol was 5α-cholestan-3β-ol (cholestanol). Stanol/stenol ratios (C$_{27}$) increased from 0.7 in the 0–1 cm sample to 1.9 in the 10–12 cm sample. Tetrahymanol peaked at 3–4 cm (188 µg g$^{-1}$ C$_{org}$) and decreased to 38 µg g$^{-1}$ C$_{org}$ in the 10–12 cm sample (S4 Table).

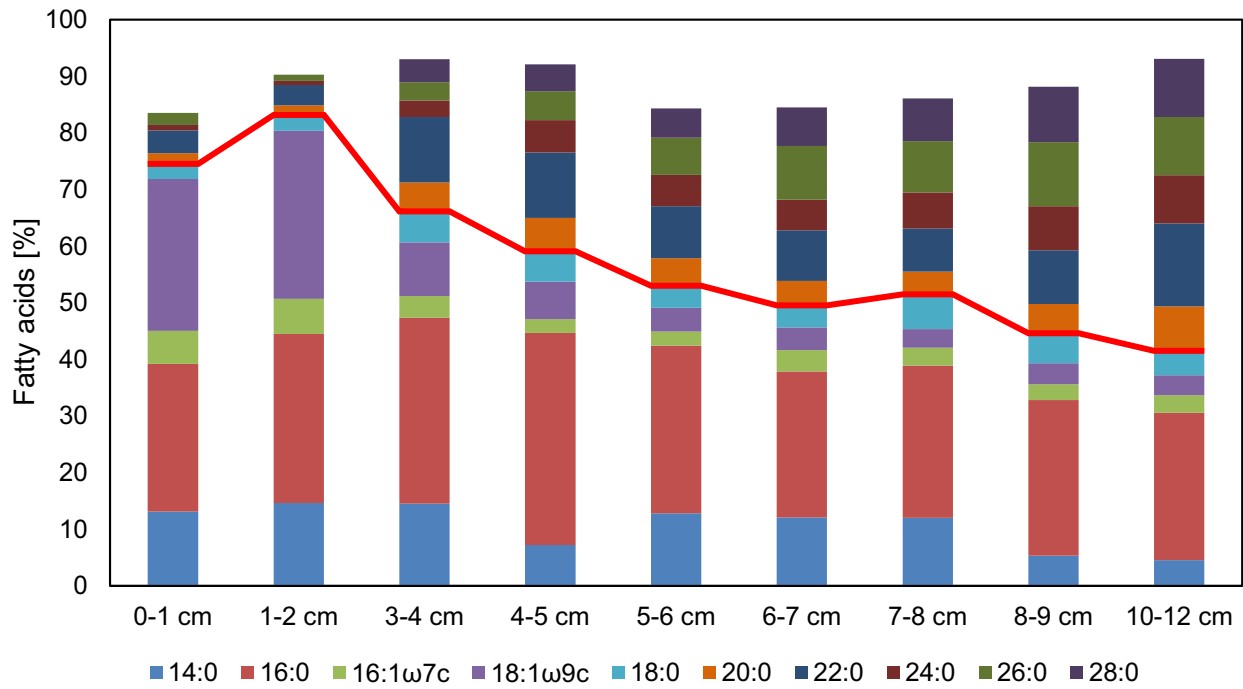

**Fig 6. Major fatty acids (FA) in the surface sediments.** Only compounds making up > 5% of the total in at least one sample are shown. The red line distinguishes short-chain FA (below) from long-chain FA. Note the relative increase of long-chain FA with sediment depth. See S2 Table for detailed values.

## 3.5. Other compounds (acyclic alcohols, hydrocarbons)

**3.5.1. N-alcohols and hydrocarbons in the water column.** The concentrations of acyclic alcohols (> 1% of the total amount of alcohols) are given in S5 Table. N-alcohols are most abundant by far in the zooplankton samples (60–90 m, 166 mg g$^{-1}$ C$_{org}$; 25-60 m, 13 mg g$^{-1}$

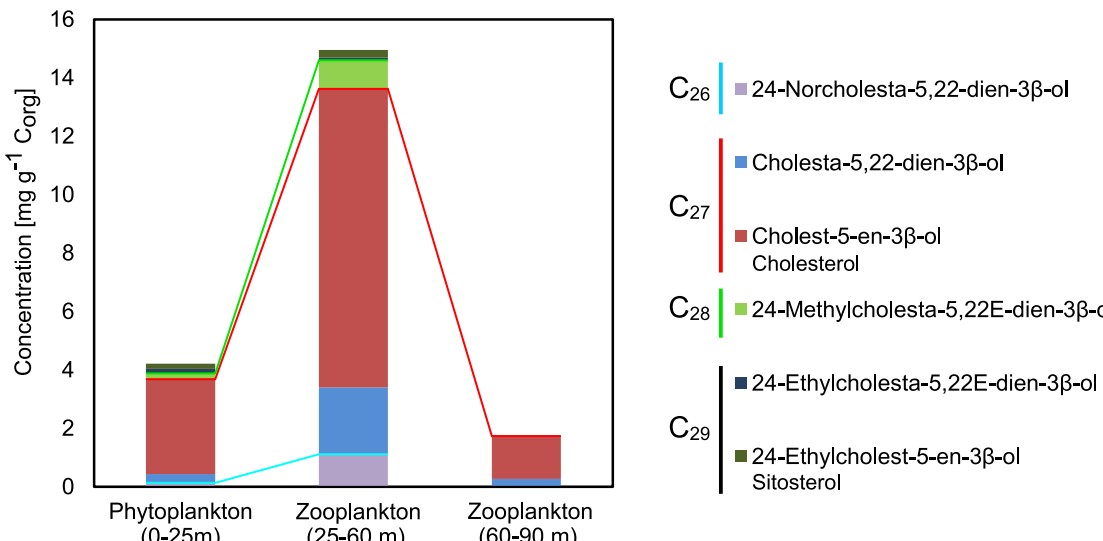

**Fig 7. Concentrations of 4-desmethylsterols in the water column samples.** Note the abundance of cholesterol (cholest-5en-3β-ol). Red and green lines delimit C$_{27}$-, C$_{28}$-, and C$_{29}$ sterols. See S3 Table for detailed values.

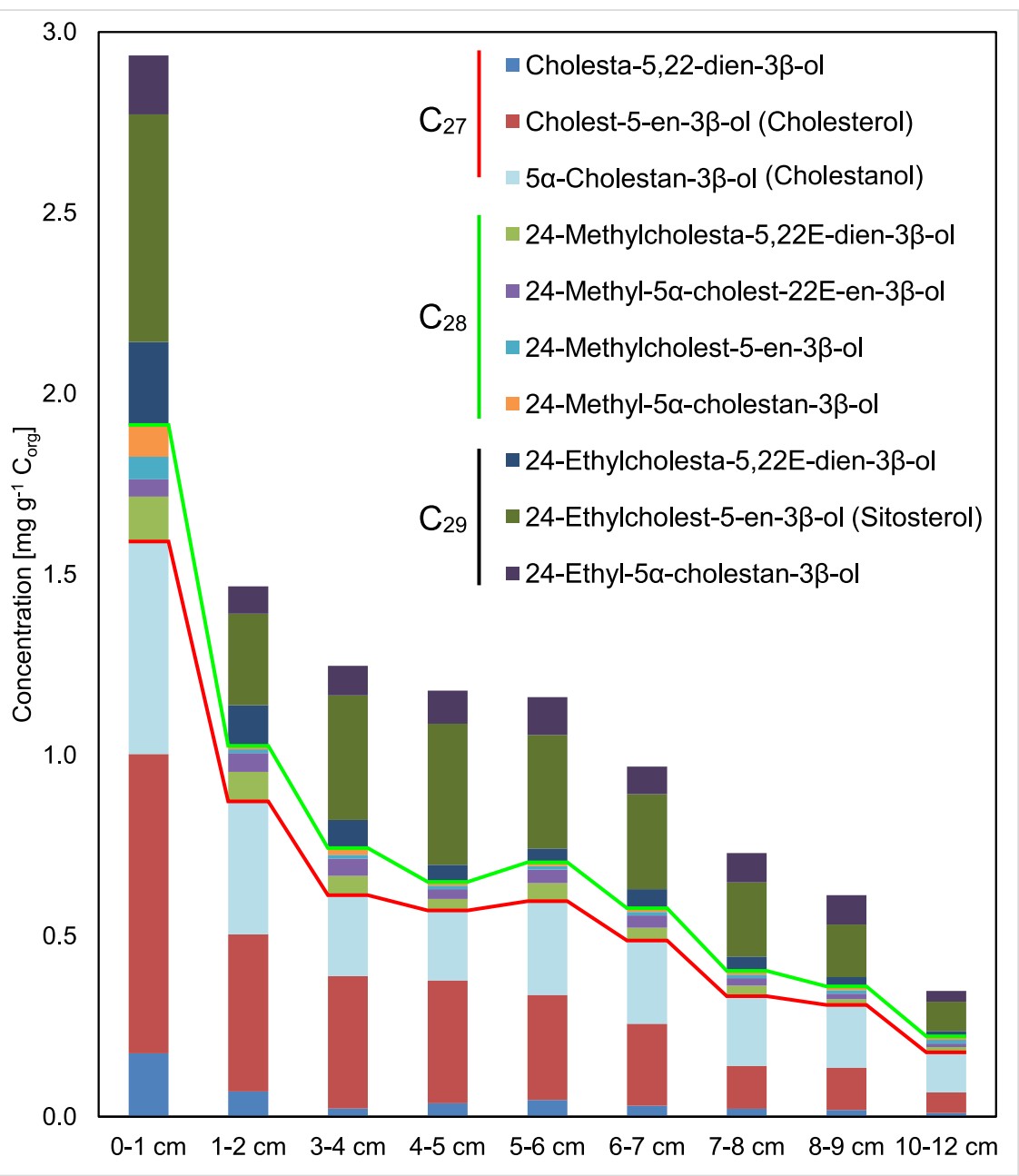

**Fig 8. Concentrations of 4-desmethylsterols in the surface sediments.** Red and green lines delimit $C_{26}$-, $C_{27}$-, $C_{28}$-, and $C_{29}$ sterols, respectively. Note that 4α,23,24-trimethyl-5α-cholest-22E-en-3β-ol (dinosterol) is not included. See S4 Table for detailed concentration data.

$C_{org}$), and 14:0 and 16:0 are the major homologues. In the filter (60–95 m) and phytoplankton (0-25 m) samples, 18:0, 20:0, and 24:0 occur in much higher *relative* abundance but the *absolute* concentrations of n-alcohols are 2–3 orders lower than in the zooplankton samples.

The phytoplankton (0-25 m) sample furthermore shows high abundances of *n*-heptadecane and 7-methylheptadecane (~ 0.9 mg g$^{-1}$ $C_{org}$) whereas hydrocarbons are virtually lacking in the zooplankton samples.

**3.5.2. N-alcohols and hydrocarbons in the sediments.** The concentrations of sedimentary alcohols are given in S6 Table. These compounds initially increased from the surface (0–1 cm, 1.4 mg g$^{-1}$ C$_{org}$) towards depth (3–4 cm, 2.9 mg g$^{-1}$ C$_{org}$). Below 6 cm, however, they rapidly decreased to nearly constant amounts of about 1 mg g$^{-1}$ C$_{org}$. In all sediments, n-docosanol (22:0) is the most abundant homologue, and a general preference of long-chain n-alcohols can be seen which notably differs from the water column distributions (see 3.3.1).

The *n*-alkane concentrations peaked at 1–2 cm and 4–5 cm. Below, these compounds declined rapidly. 6- and 7-methylheptadecanes, though very abundant in the phytoplankton sample 0–25 m (496 µg g$^{-1}$ C$_{org}$), occurred at by orders lower abundance in the topmost sediment (0–1 cm; 10.6 µg g$^{-1}$ C$_{org}$) and below (<1 µg g$^{-1}$ C$_{org}$).

# 4. Discussion

Lipids in the water column of the EGB derive from the organism community producing and transforming biomolecules *in situ*, and from allochthonous sources (e.g. terrestrial plants). Some key compounds and their putative origins are given in Table 1, and will be discussed in the following.

## 4.1. Origin and fate of FA

**4.1.1. 18:1ω9 (oleic acid): phototrophic or heterotrophic sources?** The most abundant unsaturated FA in the EGB samples studied, 18:1ω9, has various possible sources. In productive surface waters, cyanobacteria and algae are probably the main producers [26–28]. In addition, 18:1ω9 is a main FA in many ciliates and its presence in copepod lipids has therefore been interpreted as a marker for feeding on heterotrophs [36]. Contributions from heterotrophic microorganisms thriving at the chemocline may plausibly explain the presence of 18:1ω9 as the single most abundant unsaturated FA in the filter sample (60–95 m). 18:1ω9 is even more abundant in the zooplankton sample (60–90 m) from the same depth, suggesting a selective feeding behaviour of the dominant deep-water copepod, *Pseudocalanus* spp., on food sources rich in 18:1ω9, particularly ciliates. However, in copepods, synthesis of 18:1ω9 *de novo* or by conversion of 18:0 may also occur [29].

**4.1.2. Polyunsaturated FA: Markers for primary producers in the water column.** Polyunsaturated C$_{18}$ FA (18:2ω6, 18:3ω3), making up 15% of the total FA in the phytoplankton (0–25 m), are typically produced by algae including cyanobacteria [26–28, 37]. Whereas these compounds are also abundant in the shallow zooplankton sample (25-60 m, ~15%), their absence (18:3ω3) or low abundance (18:2ω6) in the deeper samples suggests that they are immediately recycled in the uppermost water column of the EGB (Fig 5).

**Table 1. Key lipids of the water column and their putative main sources.**

| Component | Source | References |
|---|---|---|
| 18:1ω9 | ciliates, algae, cyanobacteria | [25–29] |
| 18:2ω6, 18:3ω3 and- ω6 | cyanobacteria | [26, 28] |
| 20:5ω3 | diatoms | [30] |
| 22:6ω3 | dinoflagellates | [30] |
| wax alcohols | copepods | [31, 32] |
| n-heptadecanes, methylheptadecanes | cyanobacteria | [26, 33] |
| C$_{27}$ sterols | zooplankton, algae | [34, 35] |
| C$_{28}$ sterols | algae | [34, 35] |
| C$_{29}$ sterols | algae, higher terrestrial plants | [34, 35] |

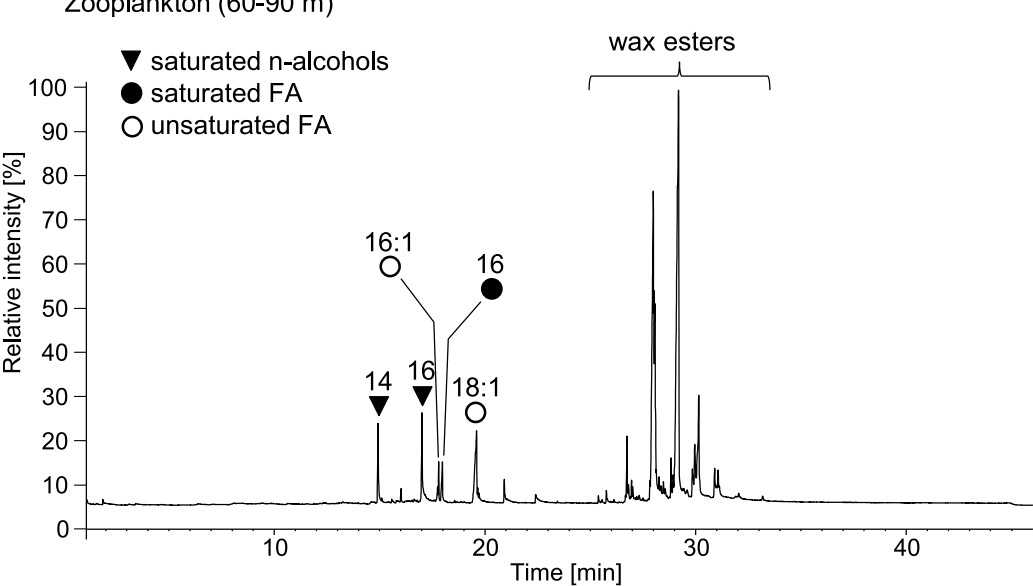

**Fig 9. Thermodesorption-GC/MS chromatogram (0.8 mg, 360˚C, 30 s) of the zooplankton (60–90 m) sample dominated by *Pseudocalanus* spp.** Note the high amounts of wax esters released; these compounds are comprised of 14:0 and 16:0 n-alcohols esterified to the 18:1ω9c fatty acid (FA) and other minor FA. See S1 Text for description of method.

The polyunsaturated FA 20:5ω3 and 22:6ω3 have been associated with diatoms and dinoflagellates, respectively [30]. Feeding of copepods on these algae plausibly explains the presence of these FA in the zooplankton samples. Concentrations of 20:5ω3 and 22:6ω3 are strongly decreased in the sediment samples, pointing at a rapid recycling of these compounds in the water column (Fig 5).

**4.1.3. Fate of FA in the sediments.** Similar to the samples from the upper oxygen-depleted zone, 18:1ω9 is the most abundant unsaturated FA in the topmost sediment of the EGB, whereas typical 'phototrophic' FA such as 18:3ω3, 20:5ω3, and 22:6ω3 are absent, or occur at very low abundance. The FA distribution in the topmost sediments at station TF0271 of the EGB thus represents the heterotrophic decomposers, including deep-water zooplankton, rather than the phototrophic primary producers in the upper mixed layer.

In sediments, individual FA degradation rates critically depend on the carbon chain lengths and the degree of unsaturation [38–40]. This explains the strong relative decrease of 18:1ω9 as well as the increasing predominance of long-chain saturated FA with sediment depth (Fig 6). Most of these long-chain FA likely originate from land plant matter [41, 42] that has been introduced into the EGB via rivers and currents, or by aeolian transport [cf. 43]. Our observations are in good agreement with the hypothesis that such land plant derived long-chain FA are preferentially preserved as they are embedded into microbially inaccessible matrices [38].

## 4.2. Origin and fate of wax esters and hydrocarbons

**4.2.1. Wax esters are mainly sourced by deep-water copepods.** N-alcohols in our samples probably originate from the hydrolytic cleavage of wax esters during analysis. Wax esters serve as protective coatings for example on leaf surfaces, but they also provide metabolic energy reserves [31] and buoyancy for marine copepods [44]. In the zooplankton samples, short-chain saturated n-alcohols (mainly 14:0, 16:0) occur at high abundances, especially in the zooplankton (60-90 m) sample dominated by *Pseudocalanus* spp. To verify a wax ester origin of these compounds we subjected sub-mg amounts of the untreated sample to direct

thermodesorption-GC/MS. This clearly indicated the presence of vast amounts of wax esters that mainly consisted of 18:1ω9 FA ester-bound to 14:0 and 16:0 n-alcohols (Fig 9). Therefore, the short-chain n-alcohols observed in the EGB samples after analytical hydrolysis can be regarded as biomarkers for copepod storage lipids.

**4.2.2. Do sedimentary n-FA derive from wax ester n-alcohols?**    Even in the topmost sediments, n-alcohols were only present at low concentrations, indicating that copepod-derived wax esters are degraded before they can be incorporated into the sediments. At second glance, however, an intriguing observation is that the wax ester derived distribution of 18:1ω9 FA and short-chain n-alcohol homologues (14:0, 16:0) in the zooplankton (60–90 m) is quite exactly mirrored in the distribution of FA homologues in the topmost sediment (Fig 10). Likewise, the concentrations of 14:0 FA are several times higher in the surface sediments as in the water column where instead the 14:0 n-alcohol is abundant (compare S1 Table, S2 Table and S5 Table), thus suggesting a direct link between these compounds. In the sediment, the resulting short-chain FA are then subject to degradation. The growing predominance of 16:0 FA with sediment depth (Fig 10) can be plausibly explained by a more rapid degradation of both 14:0 and 18:1ω9 FA due to shorter chain length and unsaturation, respectively (as discussed above).

We therefore posit that the input and taphonomy of zooplankton-derived storage lipids may, temporarily at least, represent an important control on FA distributions in the surface sediments of the EGB. A predominance of the 16:0 FA is typically observed in modern and ancient sediments and has traditionally been attributed to inputs of microbial FA, or microbial reworking of algal FA, in the sediment (see [45] for a review). Our observations from the EGB suggest that (i) decomposition of copepod wax esters, (ii) transformation of the resulting n-alcohols into the corresponding FA, and (iii) subsequent biodegradation of the resulting FA may comprise an alternative route leading to the 'classical' sedimentary FA distribution. It can be anticipated that these processes are mediated by sedimentary or bottom water microorganisms thriving under suboxic or anoxic conditions, but this needs to be verified by further investigations.

Enhanced relative abundances of long-chain n-alcohols in the deeper sediment layers again indicate contributions from terrestrial sources. The relative enrichment of these compounds with depth may plausibly result from their higher stability against degradation as compared to the short-chain homologues.

**4.2.3. 6- and 7-methylalkanes as biomarkers for surface water cyanobacteria in sediments.**    Linear and mid-chain methylated heptadecanes are classical biomarkers of cyanobacteria. These compounds are synthesized by enzymatic transformation of $C_{18}$ FA [36]. In *Nodularia spumigena*, n-octadecanoic acid (18:0 FA) is first transformed into an aldehyde, and then converted to n-heptadecane [33]. This compound is the most common alkane in cyanobacteria (including Nodularia spp.; [26]), which is in good agreement with its high abundance in the phytoplankton (0-25 m) sample (496 mg $g^{-1}$ $C_{org}$). Mid-chain branched alkanes, namely 6-, 7-, and 8-methylheptadecanes, derive from a methyltransferase reaction from a 18:1 FA precursor [33] and are even more specific for cyanobacterial inputs to sediments [46]. Given that cyanobacterial blooms have been frequently reported in the EGB over the last decades [47] we expected to find these biomarkers abundant in the shallow sediments. However, concentrations of 6- and 7-methylheptadecanes in the sediments are are by two to three orders lower as in the particulate matter of the surface water (μg- *vs*. mg $g^{-1}$ $C_{org}$ range). When compared to FA, whose concentrations kept roughly in the same order in both pools (mg $g^{-1}$ $C_{org}$ range; S1, S2), these hydrocarbons have been relatively depleted either in the water column or immediately after deposition in the topmost sediments. Nonetheless, recent work showed 6- and 7-methylheptadecanes keeping at concentrations between 0–2 μg $g^{-1}$ $C_{org}$ even in deeper layers of the EGB sediments. It has been proposed that these compounds can be used as tracers for the reconstruction of $N_2$-fixing cyanobacterial blooms in the Baltic Sea [48].

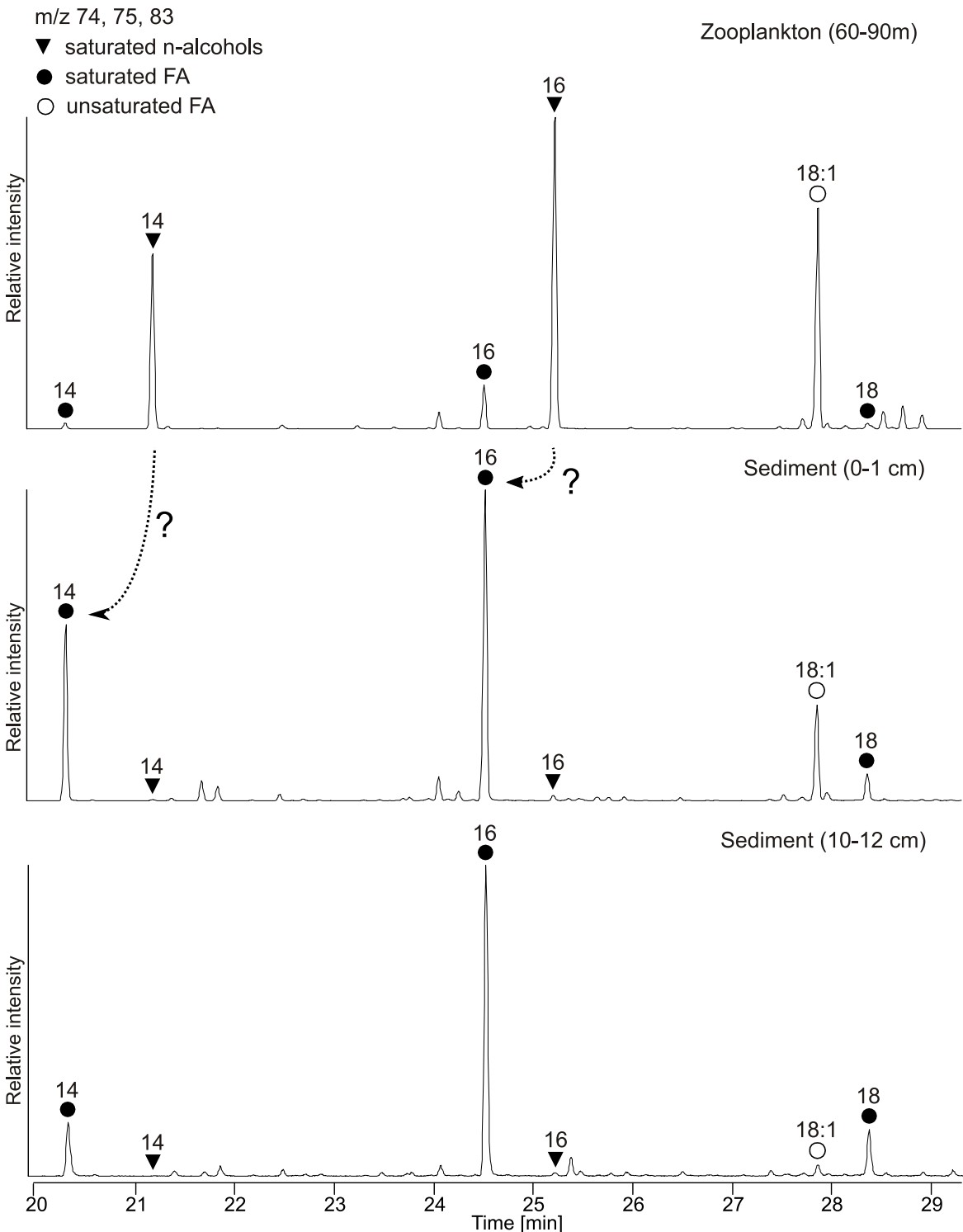

**Fig 10. Partial ion chromatograms (m/z 74, 75, 83) showing the distributions of short chain n-alcohols (as TMS-derivatives) and fatty acids (FA, as methyl ester derivatives) in the Zooplankton (60–90 m) sample and selected sediment samples.** Note the congruency of n-alcohols in the zooplankton (60–90 m) and the FA in the topmost sediment (0–1 cm) samples; dashed arrows illustrate the proposed transformation of n-alcohols into the corresponding n-FA, see text for discussion. FA in the deeper sediment (10–12 cm) reveal a relative decrease of 14:0 and 18:1ω9c and an enrichment of saturated FA of greater chain length due to biodegradation.

### 4.3. Origin and fate of sterols

**4.3.1. Enigmatic occurrence of cholesterol in the cyanobacterial bloom.** The carbon number of sterols provides information about the biological source [34]. $C_{27}$ sterols (cholesteroids) are the major sterols in zooplankton, but these compounds are also present in many algae, including dinoflagellates and diatoms [49]. $C_{28}$ sterols occur predominantly in algae, including diatoms [34, 35]. $C_{29}$ sterols indicate an input of terrigeneous organic matter, with sitosterol being the main vascular plant sterol, and some algae (many diatoms and green algae) [34, 35].

The high abundance of cholesterol (3,3 mg $g^{-1}$ $C_{org}$) in the phytoplankton sample (0-25 m) is surprising, given that the sample contains nearly exclusively cyanobacteria and virtually no copepods (according to microscopic inspection, see 3.2). Bacteria in general are not regarded as producers of cholesterol and other 4-desmethylsterols. However, reports on $C_{27}$ –$C_{29}$ sterols (including cholesterol) in cyanobacteria exist, but these are controversial [35 and references therein]. Hence, the source of cholesterol in the upper mixed layer remains enigmatic. Possibly, microzooplankton, fecal pellets or lipid droplets released by dead copepods may have contributed this compound but this needs further investigations.

**4.3.2. Diagenetic bias favouring terrestrial $C_{29}$ sterols?** Like in the water column, cholesterol is the most abundant sterol in the topmost sediments at station TF0271. This is well in line with the high abundance of mesozooplankton and the fact that copepod excretes can contain up to 4 ng cholesterol per pellet [50, 51]. However, the amount of cholesterol decreases strongly with sediment depth. At the same time, the sitosterol/cholesterol ratio increases (0–1 cm, 0.76; 10–12 cm, 1.42), which may suggest either (i) a higher relative input of sitosterol in the past or (ii) that the partly 'terrestrial' derived sitosterol is more resistant against sedimentary degradation processes (Fig 8). The latter assumption is supported by the stanol/sterol ratios that can be used as a marker for anaerobic microbial transformations [52]. The trend to higher stanol/sterol ratios for the $C_{27}$ sterols than for the $C_{29}$-sterols (S4 Table) support the idea of a more intense biodegradation of planktonic OM ($C_{27}$-sterols) as compared to terrestrial OM ($C_{29}$-sterols). One explanation for this phenomenon can be found in their different nature and modes of sedimentation. Terrestrial plant material had already undergone strong abiotic and biotic degradation in soils and during long term transport to sites further offshore [53]. After deposition, sedimentary bacteria may prefer the more easily accessible planktonic organic matter over the more recalcitrant terrestrial material.

## 5. Conclusion

Biomarkers are often interpreted as direct snapshots of past environments, but their degradation and sedimentary processes may considerably impact their distributions. Our observations at station TF0271 in the stratified Eastern Gotland Basin (EGB) indicate that primary produced particulate OM is heavily modified by mesozooplankton grazing. This overprint adds on the influence of heterotrophic microorganisms, together resulting in a major decoupling of the biomarker signals from the productive upper mixed layer and the oxygen-depleted bottom waters and sediments. Lipid biomarkers in the topmost sediments of the EGB mostly reflect heterotrophic decomposers, namely deep-water zooplankton, rather than important phototrophic primary producers such as cyanobacteria. In addition to sterols and fatty acids (FA), wax esters originating from copepods are major compounds particularly in the deeper water layers. Whereas these wax esters were not found preserved intact even in the topmost deposits, their n-alkyl constituents strongly contribute to the lipid inventory of the surface sediments. Our data indicate that the taphonomy of wax esters is associated with the transformation of their component n-alcohols (14:0 and 16:0) to the corresponding FA which are further

degraded according to known pathways. Meanwhile, the relative abundances of long-chain saturated FA, -n-alcohols and $C_{29}$ sterols (namely sitosterol) continuously increase with sediment depth. This may not only be explained by a greater terrestrial OM input in the past. It may also reflect the more refractory nature of these 'left-overs' of preceding harsh transport and degradation processes, resulting in additional blurring of the primary autochthonous biomarker signal in the EGB sediments.

## Supporting information

**S1 Table. Concentrations of individual fatty acids (FA) in the water column.** Only compounds making up > 1% of the total FA in at least one of the samples are shown. Numbers denote carbon numbers (= chain length) of n-FA and the number of double bonds, respectively (e.g. 18:1 represents n-octadecenoic acid). Numbers in superscript refer to the position and configuration of double bonds; roman numbers in superscript refer to minor isomers whose double bond positions have not been determined; *ai*-15 refers to 12-methyltetradecanoic acid (*anteiso*-pentadecanoic acid). Bars illustrate the relative abundances of individual compounds in a given fraction. No entry: compound not detected, or present in very low amounts (i.e., not quantified). *Only relative abundances (in % of the total) are available for the Filter (60–95 m) sample (values given in italics).
(PDF)

**S2 Table. Concentrations of individual fatty acids (FA) in the surface sediments.** Only compounds making up > 1% of the total FA in at least one of the samples are shown. Numbers denote carbon numbers (= chain length) of n-FA and the number of double bonds, respectively (e.g. 18:1 represents n-octadecenoic acid). Numbers in superscript refer to the position and configuration of double bonds; roman numbers in superscript refer to minor isomers whose double bond positions have not been determined; *i*-15, *ai*-15, and *i*-16 refer to 13- and 12-methyltetradecanoic acid (*iso*- and *anteiso*-pentadecanoic acid) and 14-methylpentadecanoic (*iso*-hexadecanoic acid), respectively. Bars illustrate the relative abundances of individual compounds in a given sample. No entry: compound not detected, or present in very low amounts (i.e., not quantified). Note that no separation was made between NL and PLFA for the sedimentary lipids.
(PDF)

**S3 Table. Concentrations of sterols in the water column samples.** Bars illustrate the relative abundances of individual compounds in a given sample. No entry: compound not detected, or present in very low amounts (i.e., not quantified). *Only relative abundances (in % of the total) are available for the Filter (60–95 m) sample (values given in italics).
(PDF)

**S4 Table. Concentrations of sterols and stanol/stenol ratios in the surface sediments.** Bars illustrate the relative abundances of individual compounds in a given sample. No entry: compound not detected, or present in very low amounts (i.e., not quantified). Concentrations of tetrahymanol are given for comparison.
(PDF)

**S5 Table. Absolute concentration of (acyclic) alcohols in the water column samples.** Only compounds making up > 1% of the total (acyclic) alcohols in at least one of the samples are shown. Numbers denote carbon numbers (= chain length) of n-alcohols and the number of double bonds, respectively (e.g. 18:1 represents n-octadecen-1-ol). Roman numbers in superscript refer to further isomers whose double bond positions have not been determined; *i*-15

and *ai*-17 refer to 13-methyltetradecan-1-ol (*iso*-pentadecan-1-ol) and 15-methylhexadecan-1-ol (*anteiso*-heptadecan-1-ol), respectively. Bars illustrate the relative abundances of individual compounds in a given sample. No entry: compound not detected, or present in very low amounts (i.e., not quantified). *Only relative abundances (in % of the total) are available for the Filter (60–95 m) sample (values given in italics).
(PDF)

**S6 Table. Absolute concentration of n-alcohols in the surface sediments.** Only compounds making up > 1% of the respective fraction in at least one of the samples are shown. Numbers denote carbon numbers (= chain length) of n-alcohols and numbers of double bonds (i.e., all compounds shown in the table are saturated). Bars illustrate the relative abundances of individual compounds in a given sample. No entry: compound not detected, or present in very low amounts (i.e., not quantified).
(PDF)

**S1 Text. Method description for Fig 9 (Thermodesorption-GC/MS).**
(PDF)

## Acknowledgments

We thank two anonymous reviewers and the associated editor C. Glombitza for their thoughtful comments on the manuscript. We also thank Natalie Loick-Wilde, Beate Stawiarski and Susanne Busch for their support with plankton sampling and identification of the species. We are furthermore grateful to Cornelia Conradt, Andreas Reimer and Birgit Röring for support with the sample workup and $C_{org}$ measurements. The crew of R/V *Alkor* cruise AL483 is acknowledged for their excellent collaboration during the cruise.

## Author Contributions

**Conceptualization:** Anna K. Wittenborn, Volker Thiel.

**Data curation:** Anna K. Wittenborn, Oliver Schmale.

**Funding acquisition:** Oliver Schmale, Volker Thiel.

**Investigation:** Oliver Schmale.

**Methodology:** Anna K. Wittenborn, Volker Thiel.

**Project administration:** Oliver Schmale.

**Resources:** Anna K. Wittenborn, Volker Thiel.

**Supervision:** Anna K. Wittenborn, Volker Thiel.

**Visualization:** Anna K. Wittenborn, Oliver Schmale.

**Writing – original draft:** Anna K. Wittenborn, Volker Thiel.

**Writing – review & editing:** Anna K. Wittenborn, Oliver Schmale, Volker Thiel.

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
