## [Decision Letter · Decision Letter 0]

15 Oct 2019

PONE-D-19-26525

Impact of zooplankton grazing on lipid biomarker distributions in the stratified Eastern Gotland Basin (Central Baltic Sea)

PLOS ONE

Dear Mrs Wittenborn,

Thank you for submitting your manuscript to PLOS ONE. Your mansucript has been evaluated by two independent reviewers. You will see that both reviewers have evaluated your manuscript positively. However, especially reviewer #1 has a number of major comments that I would like you to address carefully. I think that this will improove the quality of the discussion and streamline the manuscript. We feel that your article can be published in PLOS ONE provided that you address all reviewers comments. Therefore, we invite you to submit a revised version of the manuscript that addresses the points raised during the review process.

Additionally, I think the captions of the figures and in particlular of the tables in the supplmentary material should provide more details. For example, abbrevialtions in the tables should be mentioned, same as the color bars.

We would appreciate receiving your revised manuscript by Nov 29 2019 11:59PM. To enhance the reproducibility of your results, we recommend that if applicable you deposit your laboratory protocols in protocols.io, where a protocol can be assigned its own identifier (DOI) such that it can be cited independently in the future. For instructions see: http://journals.plos.org/plosone/s/submission-guidelines#loc-laboratory-protocols

We look forward to receiving your revised manuscript.

Kind regards,

Clemens Glombitza

Academic Editor

PLOS ONE

Journal Requirements:

3. Our internal editors have looked over your manuscript and determined that it may be within the scope of our Life in Extreme Environments Call for Papers. The Collection will encompass a diverse range of research articles to better understand life and biogeochemistry in extreme environments. Additional information can be found on our announcement page: https://collections.plos.org/s/extreme-environments. If you would like your manuscript to be considered for this collection, please let us know in your cover letter and we will ensure that your paper is treated as if you were responding to this call. If you would prefer to remove your manuscript from collection consideration, please specify this in the cover letter.

4.  We note that Figure 1 in your submission contains [map/satellite] images which may be copyrighted. All PLOS content is published under the Creative Commons Attribution License (CC BY 4.0), which means that the manuscript, images, and Supporting Information files will be freely available online, and any third party is permitted to access, download, copy, distribute, and use these materials in any way, even commercially, with proper attribution. For these reasons, we cannot publish previously copyrighted maps or satellite images created using proprietary data, such as Google software (Google Maps, Street View, and Earth). For more information, see our copyright guidelines: http://journals.plos.org/plosone/s/licenses-and-copyright.

Reviewers' comments:

Reviewer's Responses to Questions

**Comments to the Author**

1. Is the manuscript technically sound, and do the data support the conclusions?

Reviewer #1: Partly

Reviewer #2: Yes

2. Has the statistical analysis been performed appropriately and rigorously? 

Reviewer #1: N/A

Reviewer #2: N/A

3. Have the authors made all data underlying the findings in their manuscript fully available?

Reviewer #1: Yes

Reviewer #2: Yes

4. Is the manuscript presented in an intelligible fashion and written in standard English?

Reviewer #1: Yes

Reviewer #2: Yes

5. Review Comments to the Author

Reviewer #1: While the data quality is good and the manuscript well written, I found some major weak points in their arguments. The manuscript would require major revision before acceptance for publication.

Major comments

1. The author did not present their abundance data (both organisms and carbon content) relative to per unit volume of water.

- On Line 27, they wrote “very high cell counts…”. Cell counts are typically presented in the unit of individuals per unit volume of sample. I did not see any such data in the manuscript; the data presented were all relative proportions.

- Such data, including the abundance of phytoplankton and zooplankton per unit volume of water or the particulate organic carbon (POC) content per unit volume of water, are useful because these are the types of data commonly reported in oceanographic literature. Such information would allow the reader to place the results in a broader environmental context.

- While the discussion on the relationship between 14:0 n-alcohol and 14:0 FA is interesting (Line 358-380), I would like to see quantitative assessment that takes the zooplankton lipid export and sediment lipid influx into account. Ideally, such calculation should make use of water-column particle or POC flux data and sedimentation rate data. I do not think their comparison based on μg lipid/g Corg is correct, as most of the zooplankton biomass (Corg) is possibly degraded in the water column.

2. Their title “Impact of zooplankton grazing on lipid biomarker distributions in the stratified Eastern Gotland Basin” is misleading.

- When seeing such a title, most of the reader would expect a process-oriented study that examines which lipids are consumed or transformed by zooplankton digestion, which stay unaltered, and what might be the enrichment or depletion factors for different lipids.

- Discussion in the aforementioned direction was inadequate in the manuscript. Their data did show that labile cyanobacterial markers can be found in the shallow but not the deep zooplankton sample, indicating cycling of these markers (mainly polyunsaturated FA) in the upper water column. However, for the markers that are present both in the phytoplankton and surface sediment sample but not the zooplankton samples (i.e., 7-methylheptadecane), some further explanation is needed. This is crucial in that methylheptadecanes are “useful” biomarkers for paleoceanographic studies. I wonder how much (weight) zooplankton material was used for lipid extraction; perhaps a higher amount of material could improve the detection of methylheptadecanes in the samples.

- The same issue actually applies to those terrestrial biomarkers. These biomarkers are abundant in the surface sediment – is zooplankton grazing involved in the enrichment (active transport)? Or do these lipids settle together with other clastic material (passive transport), without passing the guts of the zooplankton? Again, I wonder if a higher amount of extraction material will help the authors to detect these compounds in zooplankton samples.

- I also hope that the authors can reorganize their discussion to make the manuscript more interesting. Instead of discussing different lipid groups, they can talk about different processes, such as “autochtonous lipids recycled in the water column”, “autochtonous lipids reaching seafloor via active transport”, “autochtonous lipids reaching seafloor via active transport”, etc.

- If the authors have no intention to add additional data to match their title, I suggest that they should use a different title.

Minor comments

L35: “time scale of decades”: I did not see any description/discussion related to time scale in the main text

L66: EGB: spell out the name

L77: Remove “;”

L113-119: Specify the mesh size of the Aptein-net

L130-134: Specify the filtered water volume. I wonder if an inadequate volume of filtered water resulted in the low lipid diversity on the filter

L145: Specify the weight of samples used for extraction

L185: The Chl a content in the upper mixed layer did not seem to reach 1.5 μg/L

L188: To me the salinity seems to reach 7 PSU

L214, L275 (and other places): “57 % PLFA”: percentage of what? “> 1 % of the total”: total of what?

L218: “the latter samples”: specify which samples are referred to

L231: replace “except” with “but not”

L238-239: What is the relationship between the statements “no separation between NL and PLFA was made” and “it can be expected that degradation processes and (proto-) kerogen formation would have influenced the biological distributions”?

L244-245: Change “… it was much less abundant (< 5 %) in the water column samples” to “… its proportion was much lower (< 5 %) in …”

L265: Which scientific name in the preceding sentence does “sitosterol” refer to?

L290-291: Report the concentration of 7-methylheptadecane in the topmost sediment

L314: Add “,” after “…(~15%)”

L320: Remove “still”

L322-327: The issue of 18:1ω9: Isotopic measurements of this lipid may help to quantify the relative contribution of different sources and reduce the level of speculation

L358: Change “amounts” to “concentrations”

L385: This does not seem to be a good header. Should we expect to see “surface water cyanobacteria” in sediment?

L421: Do you mean Fig. 8 (instead of Fig. 6)?

L427-432: The argument here seems somehow disconnected from the preceding text. The cited paper [49] is an arctic research; is the result applicable to other area? The sentence “phytoplankton and zooplankton are mainly degraded during summer by photodegradation in the upper water column” sounds counter-intuitive; didn’t the authors just show us rapid biological lipid cycling in the upper water column?

Fig. 5 (and supplementary tables): Specify the sampling depth of Filter sample

Fig. 7: Delimit C27, C28… sterols (in analogy to Fig. 8)

Fig. 7, 8: Provide common name of the sterols (if applicable) for easier cross-referencing in the text

Fig. 8: Sterol/tetrahymanol data of the depth 3-4 cm look strange. Why did the major lipid 24-ethylcholest-5-en-3β-ol disappear in this layer?

Table S1, S3, S5: Values of the Filter sample are unclear

Table S3, S4: Provide common name of the sterols (if applicable) for easier cross-referencing in the text

Reviewer #2: The authors present data of lipid distributions in the Eastern Gotland Basin and study how these biomarker signals are transformed during/after deposition in this stratified and partly anoxic environment. For this purpose, samples enriched in the main biomass producers phyto- and zooplankton were analyzed for the lipids fatty acids, n-alcohols, and sterols, and compared to samples from the underlying sediment. In contrast to the expectation that the lipids of primary producers are predominantly found in surface sediments deposited under (partly) anoxic conditions, the data instead show that transformed remnants of wax esters derived from zooplankton are the dominating biomarkers. Deeper sediment layers are mostly comprised of biomarkers from terrestrial plants, indicating that grazing of zooplankton strongly changes the composition of the primary producer-derived lipids during and after deposition, leading to unexpected results.

The manuscript is well-written and all experiments are described in sufficient detail. Figures and Tables follow a clear layout and all data is made available in the Supplementary Information. Previous literature is fairly treated and the experimental data support the conclusions that the authors draw. I do not have any major criticism of any kind, this was a pleasure to read – well done!

Detailed comments

L66: introduce EGB here, readers who have not read the abstract first will not know the abbreviation

L120: typo: “homogenous”

L148-154: Was the efficiency/recovery of the column chromatography tested? I suggest to rename the “phospholipid fraction” into “polar lipid fraction” to also account for other types of polar lipids (glyco, amino, etc.) which might be eluted (cf. Heinzelmann et al., 2014, AEM; “Critical Assessment of Glyco- and Phospholipid Separation by Using Silica Chromatography”).

L222: perhaps change to „virtually absent”

L265: check assignment of “sitosterol” to chemical formula

Table S4: use decimal point in stanol/stenol ratios

6. PLOS authors have the option to publish the peer review history of their article (what does this mean?). If published, this will include your full peer review and any attached files.

Reviewer #1: No

Reviewer #2: No

---

## [Author Response · Author response to Decision Letter 0]

12 May 2020

Please see the attached file Response to reviewers. 

Editor

1. Additionally, I think the captions of the figures and in particlular of the tables in the supplmentary material should provide more details. For example, abbreviations in the tables should be mentioned, same as the color bars.

Done. 

File naming is correct now.

The field site access was granted by Latvian Ministry of Foreign Affairs. The permit number is ZD16AK0020.

4. Our internal editors have looked over your manuscript and determined that it may be within the scope of our Life in Extreme Environments Call for Papers. The Collection will encompass a diverse range of research articles to better understand life and biogeochemistry in extreme environments. Additional information can be found on our announcement page: https://collections.plos.org/s/extreme-environments. If you would like your manuscript to be considered for this collection, please let us know in your cover letter and we will ensure that your paper is treated as if you were responding to this call. If you would prefer to remove your manuscript from collection consideration, please specify this in the cover letter.

Thank you for consideration but we feel that our topic is less about life in extreme environments.

5. We note that Figure 1 in your submission contains [map/satellite] images which may be copyrighted. All PLOS content is published under the Creative Commons Attribution License (CC BY 4.0), which means that the manuscript, images, and Supporting Information files will be freely available online, and any third party is permitted to access, download, copy, distribute, and use these materials in any way, even commercially, with proper attribution. For these reasons, we cannot publish previously copyrighted maps or satellite images created using proprietary data, such as Google software (Google Maps, Street View, and Earth). For more information, see our copyright guidelines: http://journals.plos.org/plosone/s/licenses-and-copyright.

The map was created in our institute based on the data T. Seifert, F. Tauber, B. Kayser: 2001: "A high resolution spherical grid topography of the Baltic Sea – 2nd edition", Baltic Sea Science Congress, Stockholm 25-29. November 2001, Poster #147, https://www.io-warnemuende.de/topography-of-the-baltic-sea.html and modified for the isobaths in ArcView 10 software. 

Reviewer 1 

While the data quality is good and the manuscript well written, I found some major weak points in their arguments. The manuscript would require major revision before acceptance for publication.

Major comments

1. The author did not present their abundance data (both organisms and carbon content) relative to per unit volume of water.

- On Line 27, they wrote “very high cell counts…”. Cell counts are typically presented in the unit of individuals per unit volume of sample. I did not see any such data in the manuscript; the data presented were all relative proportions.

We agree. The amount of individuals per unit volume was not recorded during our sampling; the term ‘cell counts’ is thus misleading and has now been omitted. Instead we provide mean biomass abundance data obtained in the EGB at the time of our sampling by the IOW monitoring (N. spumigena, 122µg/l). The respective IOW report (available open access) is cited accordingly (Wasmund et al., 2017).

2. Such data, including the abundance of phytoplankton and zooplankton per unit volume of water or the particulate organic carbon (POC) content per unit volume of water, are useful because these are the types of data commonly reported in oceanographic literature. Such information would allow the reader to place the results in a broader environmental context.

We agree and included recently published data from our cruise (19-25 individuals/L, Stawiarski et al., 2019) as well as data for the Mecklenburg Bay (5-15), Arkona Basin (7-15) and the adjacent Gdansk Deep (4-18) as an orientation (Wasmund et al., 2017; Dzierzbicka-Glowacka et al., 2018).

3. While the discussion on the relationship between 14:0 n-alcohol and 14:0 FA is interesting (Line 358-380), I would like to see quantitative assessment that takes the zooplankton lipid export and sediment lipid influx into account. Ideally, such calculation should make use of water-column particle or POC flux data and sedimentation rate data. I do not think their comparison based on μg lipid/g Corg is correct, as most of the zooplankton biomass (Corg) is possibly degraded in the water column.

It is difficult to respond to this comment, as no parallel flux measurements/sediment trap experiments were performed in the scope of our cruise. To obtain lipid flux data it would have needed a different project setup which we cannot offer. Given our focus on the sedimentary record of lipid biomarkers, please understand that we used µg g-1 Corg as the most commonly reported unit in organic geochemistry.

4. Their title “Impact of zooplankton grazing on lipid biomarker distributions in the stratified Eastern Gotland Basin” is misleading.

- When seeing such a title, most of the reader would expect a process-oriented study that examines which lipids are consumed or transformed by zooplankton digestion, which stay unaltered, and what might be the enrichment or depletion factors for different lipids.

We did not explicitly provide enrichment or depletion factors for individual compounds in order to keep our manuscript concise. If desired, however, these factors can be determined using the quantitative data in the supplementary tables (e.g. S1 and S2 for FA). A discussion about which lipids are consumed or transformed by zooplankton or early diagenetic processes is given in chapter 4.

To direct the attention of the reader more towards the intended comparison of water column vs. sedimentary lipids, we changed the title to “Zooplankton impact on lipid biomarkers in water column vs. surface sediments of the stratified Eastern Gotland Basin (Central Baltic Sea)”

5. Discussion in the aforementioned direction was inadequate in the manuscript. Their data did show that labile cyanobacterial markers can be found in the shallow but not the deep zooplankton sample, indicating cycling of these markers (mainly polyunsaturated FA) in the upper water column. However, for the markers that are present both in the phytoplankton and surface sediment sample but not the zooplankton samples (i.e., 7-methylheptadecane), some further explanation is needed. This is crucial in that methylheptadecanes are “useful” biomarkers for paleoceanographic studies. I wonder how much (weight) zooplankton material was used for lipid extraction; perhaps a higher amount of material could improve the detection of methylheptadecanes in the samples.

We thank the reviewer for this important comment. We probably missed these compounds in part of our deeper samples, due to their much (by orders) lower concentrations with respect to functionalized lipids. In fact, hydrocarbons play only a very negligible quantitative role in our runs and have been ‘overwhelmed’ by the much more abundant fatty acids, alcohols and sterols to whose concentrations we actually adjusted our GC/MS analyses. The robust quantitative analysis of these hydrocarbons would have required a different analytical setup, including chromatographic separation and major concentration of the resulting fractions. We therefore rephrased our text and referred to a recent study that specifically analyzed 6- and 7-methylheptadecanes from a closed by location in the EGB (including sediment trap data; Kaiser et al., 2020). In that work methylheptadecanes were reported to occur in concentrations of +\\- 1 µg g-1 Corg in the surface sediments.

6. The same issue actually applies to those terrestrial biomarkers. These biomarkers are abundant in the surface sediment – is zooplankton grazing involved in the enrichment (active transport)? Or do these lipids settle together with other clastic material (passive transport), without passing the guts of the zooplankton? Again, I wonder if a higher amount of extraction material will help the authors to detect these compounds in zooplankton samples. 

We are not quite sure whether we understood the reviewer correctly here. There is no detection problem with the major terrestrial compounds in our samples, and it is clearly revealed from our data that these compounds increase in relative abundance with sediment depth (e.g. long-chain-FA in Fig. 6; long-chain n-alcohols in Table S6). Further, it can be considered unlikely a priori that mesozooplankton would deliberately feed on terrestrial materials, as this material is much more refractory than freshly produced plankton (higher C/N). Feeding on planktonic organisms would also be in agreement with the diurnal migration behaviour of many copepods. As the reviewer suggests, it is therefore likely that the terrestrial matter is settling “together with other clastic material (passive transport), without passing the guts of the zooplankton”

7. I also hope that the authors can reorganize their discussion to make the manuscript more interesting. Instead of discussing different lipid groups, they can talk about different processes, such as “autochtonous lipids recycled in the water column”, “autochtonous lipids reaching seafloor via active transport”, “autochtonous lipids reaching seafloor via active transport”, etc.

- If the authors have no intention to add additional data to match their title, I suggest that they should use a different title.

As we feel that the perception of the discussion may vary depending on the different interests of the readers, we would prefer to keep centering on the compound classes (thus providing a clear-cut link to the results) rather than restructuring the text. In doing so, we agree that the title should be reworded to be more explicit, see our reply to comment 4 above.

Minor comments

L35: “time scale of decades”: I did not see any description/discussion related to time scale in the main text

Considered. It has now been clarified in the text (ch. 2.1) that our estimation was based on comparison with similarly laminated cores from close-by locations in the Eastern Gotland Basin that have been dated by Moros et al. 2017.

L66: EGB: spell out the name

Done

L77: Remove “;”

Done

L113-119: Specify the mesh size of the Aptein-net

Done

L130-134: Specify the filtered water volume. I wonder if an inadequate volume of filtered water resulted in the low lipid diversity on the filter

Done

L145: Specify the weight of samples used for extraction

Done 

L185: The Chl a content in the upper mixed layer did not seem to reach 1.5 μg/L

Done, changed to 1.4.

L188: To me the salinity seems to reach 7 PSU

Done

L214, L275 (and other places): “57 % PLFA”: percentage of what? “> 1 % of the total”: total of what?

Considered and rewritten.

L218: “the latter samples”: specify which samples are referred to

Done, instead of “the latter samples” planktonic samples

L231: replace “except” with “but not”

Done

L238-239: What is the relationship between the statements “no separation between NL and PLFA was made” and “it can be expected that degradation processes and (proto-) kerogen formation would have influenced the biological distributions”?

Considered, the explanation was omitted to avoid confusion.

L244-245: Change “… it was much less abundant (< 5 %) in the water column samples” to “… its proportion was much lower (< 5 %) in …”

Done

L265: Which scientific name in the preceding sentence does “sitosterol” refer to?

Done. 24-ethylcholest-5-en-3β-ol

L290-291: Report the concentration of 7-methylheptadecane in the topmost sediment

See our comment on major comment 5.

L314: Add “,” after “…(~15%)”

Done

L320: Remove “still”

Done

L322-327: The issue of 18:1ω9: Isotopic measurements of this lipid may help to quantify the relative contribution of different sources and reduce the level of speculation

We agree and will consider this comment in our future studies.

L358: Change “amounts” to “concentrations”

Done

L385: This does not seem to be a good header. Should we expect to see “surface water cyanobacteria” in sediment?

Considered, header was reworded.

L421: Do you mean Fig. 8 (instead of Fig. 6)?

Done.

L427-432: The argument here seems somehow disconnected from the preceding text. The cited paper [49] is an arctic research; is the result applicable to other area? The sentence “phytoplankton and zooplankton are mainly degraded during summer by photodegradation in the upper water column” sounds counter-intuitive; didn’t the authors just show us rapid biological lipid cycling in the upper water column?

Considered, we removed the speculations on the role of photodegradation and reworded the paragraph.

Fig. 5 (and supplementary tables): Specify the sampling depth of Filter sample

Done. 

Fig. 7: Delimit C27, C28… sterols (in analogy to Fig. 8)

Done.

Fig. 7, 8: Provide common name of the sterols (if applicable) for easier cross-referencing in the text

Done.

Fig. 8: Sterol/tetrahymanol data of the depth 3-4 cm look strange. Why did the major lipid 24-ethylcholest-5-en-3β-ol disappear in this layer?

Considered and re-analysed.

Table S1, S3, S5: Values of the Filter sample are unclear

No Corg data was obtained for the filter samples. Therefore we just showed the relative proportions in the sample.

Table S3, S4: Provide common name of the sterols (if applicable) for easier cross-referencing in the text

Done.

 

Reviewer 2

The authors present data of lipid distributions in the Eastern Gotland Basin and study how these biomarker signals are transformed during/after deposition in this stratified and partly anoxic environment. For this purpose, samples enriched in the main biomass producers phyto- and zooplankton were analyzed for the lipids fatty acids, n-alcohols, and sterols, and compared to samples from the underlying sediment. In contrast to the expectation that the lipids of primary producers are predominantly found in surface sediments deposited under (partly) anoxic conditions, the data instead show that transformed remnants of wax esters derived from zooplankton are the dominating biomarkers. Deeper sediment layers are mostly comprised of biomarkers from terrestrial plants, indicating that grazing of zooplankton strongly changes the composition of the primary producer-derived lipids during and after deposition, leading to unexpected results.

The manuscript is well-written and all experiments are described in sufficient detail. Figures and Tables follow a clear layout and all data is made available in the Supplementary Information. Previous literature is fairly treated and the experimental data support the conclusions that the authors draw. I do not have any major criticism of any kind, this was a pleasure to read – well done!

Detailed comments

L66: introduce EGB here, readers who have not read the abstract first will not know the abbreviation

Done

L120: typo: “homogenous”

Done.

L148-154: Was the efficiency/recovery of the column chromatography tested? I suggest to rename the “phospholipid fraction” into “polar lipid fraction” to also account for other types of polar lipids (glyco, amino, etc.) which might be eluted (cf. Heinzelmann et al., 2014, AEM; “Critical Assessment of Glyco- and Phospholipid Separation by Using Silica Chromatography”).

No, it was not tested. 

Done, renamed to polar lipid fraction. 

L222: perhaps change to „virtually absent”

Done

L265: check assignment of “sitosterol” to chemical formula

Done.

Table S4: use decimal point in stanol/stenol ratios

Done.

Literature (newly included)

Dzierzbicka-Glowacka, Lidia, et al. "Seasonal changes in the abundance and biomass of copepods in the south-eastern Baltic Sea in 2010 and 2011." PeerJ 6 (2018).

Kaiser, J., Wasmund, N., Kahru, M., Wittenborn, A. K., Hansen, R., Häusler, K., ... & Arz, H. W. Reconstructing N2-fixing cyanobacterial blooms in the Baltic Sea beyond observations using 6-and 7-methylheptadecanes in sediments as specific biomarkers.

https://www.biogeosciences-discuss.net/bg-2019-455/bg-2019-455.pdf

Moros, M., Andersen, T. J., Schulz‐Bull, D., Häusler, K., Bunke, D., Snowball, I., ... & Hand, I. (2017). Towards an event stratigraphy for Baltic Sea sediments deposited since AD 1900: approaches and challenges. Boreas, 46(1), 129-142.

Wasmund, Norbert, et al. "Biological assessment of the Baltic Sea 2016." Meereswiss. Ber., Warnemünde 105 (2017).

https://www.io-warnemuende.de/tl_files/forschung/meereswissenschaftliche/berichte/mebe105_2017_assessment-bio.pdf

---

## [Editor Report · Decision Letter 1]

20 May 2020

Zooplankton impact on lipid biomarkers in water column vs. surface sediments of the stratified eastern Gotland Bain (Central Baltic Sea)

PONE-D-19-26525R1

Dear Dr. Wittenborn,

We are pleased to inform you that your manuscript has been judged scientifically suitable for publication and will be formally accepted for publication once it complies with all outstanding technical requirements.

With kind regards,

Clemens Glombitza

Academic Editor

PLOS ONE

---

## [Editor Report · Acceptance letter]

28 May 2020

PONE-D-19-26525R1 

Zooplankton impact on lipid biomarkers in water column vs. surface sediments of the stratified eastern Gotland Bain (Central Baltic Sea) 

Dear Dr. Wittenborn:

I am pleased to inform you that your manuscript has been deemed suitable for publication in PLOS ONE. Congratulations! Your manuscript is now with our production department. 

With kind regards,

on behalf of

Dr. Clemens Glombitza 

Academic Editor

PLOS ONE